# Characterizing and Targeting Genes Regulated by Transcription Factor MYBL2 in Lung Adenocarcinoma Cells

**DOI:** 10.3390/cancers14204979

**Published:** 2022-10-11

**Authors:** Yuri Lee, Zexun Wu, Seolyn Yang, Shannon M. Schreiner, Leonardo D. Gonzalez-Smith, Suhn K. Rhie

**Affiliations:** Department of Biochemistry and Molecular Medicine, Norris Comprehensive Cancer Center, Keck School of Medicine, University of Southern California, Los Angeles, CA 90033, USA

**Keywords:** MYBL2, FOXM1, lung adenocarcinoma, cell cycle, inhibitors

## Abstract

**Simple Summary:**

The clinical behavior and progression of lung cancer vary case by case. Therefore, understanding the factors that contribute to aggressive lung cancer cases is crucially needed. Previously, we showed that transcription factor MYBL2 expression is associated with the survival of lung adenocarcinoma patients. In this study, we characterized the functions of MYBL2 in lung adenocarcinoma cells by generating global binding profiles of MYBL2 and transcriptome profiles upon knockdown experiments. We discovered that MYBL2 regulates cell cycle genes by binding to the promoters of highly expressed genes in lung adenocarcinoma cells working with FOXM1. Moreover, we found that a previously reported FOXM1 inhibitor, FDI-6 (forkhead domain inhibitor 6), suppresses lung adenocarcinoma cell proliferation by inhibiting the activities of MYBL2 and FOXM1 and controlling cell death and cell cycle genes. Our findings provide valuable insights into the molecular mechanisms of aggressive lung cancer and suggest potential targets and treatments for the disease.

**Abstract:**

Overexpression of MYBL2 is associated with poor survival of lung adenocarcinoma patients, but the molecular mechanism by which it regulates transcription and carcinogenesis has not yet been elucidated. In this study, we performed ChIP-seq using an MYBL2-targeted antibody and discovered that MYBL2 primarily binds to the promoters of highly expressed genes in lung adenocarcinoma cells. Using a knockdown experiment of MYBL2 and global transcriptome profiling, we identified that over a thousand genes are dysregulated by MYBL2, and MYBL2 acts as a transcriptional activator in lung adenocarcinoma cells. Moreover, we revealed that the binding sites of FOXM1 are largely shared with MYBL2 binding sites, and genes involved in cell cycle phase transitions are regulated by these transcription factors. We furthermore investigated the effect of a previously reported FOXM1 inhibitor, FDI-6, in lung adenocarcinoma cells. We demonstrated that FDI-6 decreases the proliferation of lung adenocarcinoma cells and inhibits the activities of FOXM1 as well as MYBL2. Moreover, we found that genes involved in cell death and cell cycle are inhibited by FDI-6. Overall, our findings suggest that MYBL2 and FOXM1 activate cell cycle genes together, acting as oncogenic transcription factors in lung adenocarcinoma cells, and they are potential treatment targets for the disease.

## 1. Introduction

Lung cancer is the most common cancer worldwide and a leading cause of cancer-related deaths [1]. Despite advances in detection and therapeutic tools, most lung cancer patients have a poor prognosis, with an overall 5-year survival rate of less than 15% [2,3,4,5]. Therefore, it is imperative to understand the factors contributing to lung cancer progression to improve the diagnosis and treatment of lung cancer patients. Small cell lung cancer and non-small cell lung cancer are the two main types of lung cancer, with the latter accounting for about 85% of all cases of lung cancer cases [6]. Among non-small cell lung cancer subtypes, lung adenocarcinoma is the most common subtype. Previously, we and other research groups have studied genes that contribute to lung adenocarcinoma progression and found that overexpression of *MYBL2* (alias B-MYB) is associated with poor survival of lung adenocarcinoma patients [7,8].

MYB Proto-Oncogene Like 2 (MYBL2), which is located in chromosome 20q13, belongs to the Myb transcription factor (TF) family, and it is reported to be involved in cell survival [9]. However, the biological function of MYBL2 in cell survival is reported to vary depending on the cell type. MYBL2 has pro-survival functions that protect against cell death in multiple cell types. For example, MYBL2 is reported to regulate apoptotic-related genes in fibroblasts, T cells, and neuroblastoma cells, promoting cell survival [2,3,4,5]. MYBL2 overexpression induces anti-apoptotic Bcl-2 (B-cell lymphoma 2) expression in T cells and B cells [2,3,9]. Moreover, Yuan et al. showed that MYBL2 suppresses autophagy and promotes cell survival in ovarian oocytes [10].

On the contrary, MYBL2 is reported to have anti-survival functions in neural cells. When the cells are exposed to apoptotic stimuli such as reactive oxygen species (ROS), endoplasmic reticulum (ER) stress and DNA damage, MYBL2 accelerates neuronal apoptosis [9,11]. Moreover, the knockdown of MYBL2 protects against cell death in neural cells [12,13]. In blood cells, it is suggested that MYBL2 acts as a tumor suppressor gene contributing to myeloid malignancies. *MYBL2* expression levels in myeloid malignancy patients were less than 50% of those in healthy individuals. Moreover, reducing the expression of *MYBL2* in mice induced symptoms of myeloid malignancies in the animals [14]. Therefore, elucidating the functions and molecular mechanisms of MYBL2 in a cell-type-specific manner is crucially needed.

In this study, we characterized the molecular functions of MYBL2 in lung adenocarcinoma cells. We performed ChIP-seq (chromatin immunoprecipitation coupled with sequencing) to identify binding sites of MYBL2 in lung adenocarcinoma cells. We profiled the transcriptome upon MYBL2 knockdown to identify genes regulated by MYBL2 in lung adenocarcinoma cells. We also characterized the relationship between MYBL2 and FOXM1 by mapping FOXM1 binding profiles and investigating the transcriptome changes. Lastly, we assessed potential treatment options for the disease by treating lung adenocarcinoma cells with an inhibitor targeting MYBL2 and FOXM1.

## 2. Materials and Methods

### 2.1. Study Design

We obtained and analyzed RNA-seq data generated in lung adenocarcinoma and other normal lung-relevant cells from the Encyclopedia of DNA Elements (ENCODE) (https://www.encodeproject.org/, accessed on 4 December 2020), the Gene Expression Omnibus (GEO) (https://www.ncbi.nlm.nih.gov/geo/, accessed on 27 July 2022), and the Cancer Genome Atlas (TCGA) (https://www.cancer.gov/about-nci/organization/ccg/research/structural-genomics/tcga, accessed on 24 September 2020) databases. We generated ChIP-seq and RNA-seq data in lung adenocarcinoma cells to identify genes regulated by MYBL2 and FOXM1. To investigate the effect of inhibitors, RNA-seq was performed upon treatment with forkhead domain inhibitor 6 (FDI-6). A list of all genomic datasets used in this study can be found in Appendix A.

### 2.2. Cell Culture

Human lung adenocarcinoma A549 and NCI-H2126 cells were obtained from ATCC (Cat #CCL-185 and Cat # CCL-256, ATCC, Manassas, VA, USA). Cells were cultured in Dulbecco’s modified Eagle’s medium (DMEM) (Cat #10-013-CV, Corning, Tewksbury, MA, USA) or RPMI 1640 medium (Cat #10-040-CVR, Corning, Tewksbury, MA, USA) containing 10% fetal bovine serum (Cat #30-2022, ATCC, Manassas, VA, USA) and 1% penicillin/streptomycin (Cat #97063-708, VWR Chemicals, Radnor, PA, USA) at 37 °C in a humidified incubator with 5% CO_2_.

### 2.3. FDI-6 Treatment

FDI-6 was obtained from MedChemExpress (Cat #HY-112721, Monmouth Junction, NJ, USA). For cell proliferation analysis, A549 cells and NCI-H2126 cells were seeded in 24 wells at a density of 0.5 × 10^5^ cells/well and treated with various concentrations of FDI-6 (0, 10, 20 μM). After treating cells with FDI-6 for 24, 48, and 72 h, the cells were harvested, and live cells were counted using a hemocytometer. For the colony formation assay, cells were seeded in 12-well culture plates. After being incubated for 24 h at 37 °C, A549 and NCI-H2126 cells were treated with various concentrations of FDI-6 (0, 10, 20 μM) for 24 h. Following the 24 h FDI-6 treatment, cells were incubated for 14 days with the normal media without FDI-6. After 14 days, the colonies were fixed with cold methanol for 10 min, stained with 0.5% crystal violet for 10 min, and then washed under running tap water. Experiments were repeated at least three times. A549 cells that were treated with FDI-6 (20 μM) for 24 h were used for Western blot and RNA-seq experiments (see below sections for details).

### 2.4. Antibodies

Antibodies against beta-actin (ACTB), MYBL2, and FOXM1 were acquired from Proteintech (Cat #66009-1-lg, #18896-1-AP, #13147-1-AP, Chicago, IL, USA). Antibodies against FOXM1 and CENPA were purchased from Cell Signaling Technology (Cat #20459S, #2186S, Beverly, MA, USA).

### 2.5. Chromatin Immunoprecipitation (ChIP)

Human lung adenocarcinoma A549 cells were used to perform ChIP as previously described [15]. Briefly, A549 cells were grown to 70–90% confluence and crosslinked with 1% formaldehyde (Cat #10790-708, VWR, Radnor, PA, USA) for 10 min and quenched with 125 mM glycine (Cat #4840, MilliporeSigma, St. Louis, MA, USA) for 5 min at room temperature (RT). Cells were lysed in a cell lysis buffer using a Dounce homogenizer. Nuclei were lysed in a nuclei lysis buffer and snap-frozen using liquid nitrogen. Chromatin was sonicated to an approximate length of 200–500 bp using the Bioruptor Pico sonication device (Cat #B01060010, Diagenode, Danville, NJ, USA). Samples were then centrifuged at max speed for 10 min and incubated overnight with the target antibody. Immunoprecipitants were collected with protein A/G magnetic beads (Cat #88803, ThermoFisher Scientific, Waltham, MA, USA) for 1 h. The beads were washed and eluted with elution buffer. Chromatin was incubated at 67 °C for 16 h to reverse crosslink and purified using a Qiagen DNA purification spin column (Cat #28006, Qiagen, Hilden, Germany) and eluted with elution buffer. ChIP libraries were generated using KAPA Hyper Prep Kit (Cat #KK8505, Roche, Wilmington, MA, USA) and NEXTflex DNA Barcodes (Cat #514014, Bioo Scientific Corporation, Austin, TX, USA) as previously described (26–28). ChIP libraries were amplified using PCR and purified with AMPure XP beads (Cat #A63881, Beckman Coulter, Brea, CA, USA). The quality and concentration of libraries were determined using high-sensitivity DNA electrophoresis with a 2100 Bioanalyzer system (Agilent Technologies, Santa Clara, CA, USA) and Qubit (ThermoFisher Scientific, Waltham, MA, USA).

### 2.6. ChIP-Seq Analysis

ChIP-seq was performed using a targeted antibody, as previously described [15,16]. The ChIP-seq libraries were sequenced to have at least 20 million read pairs (Appendix A) using an Illumina NovaSeq 6000. According to the ENCODE ChIP-seq guideline [17], we generated two biological replicates of MYBL2 and FOXM1 ChIP-seq datasets that passed QC (quality control). The ENCODE3 ChIP-seq pipeline (https://github.com/ENCODE-DCC/chip-seq-pipeline, accessed on 6 August 2016) was used to analyze ChIP-seq data as previously done [18]. The raw sequencing reads were aligned to the human reference genome hg38 using BWA [19]. To determine the quality of the library, the library complexity was computed to identify quality metrics, which included the PCR bottleneck coefficient (PBC), non-redundant fraction (NRF), normalized strand cross-correlation coefficient (NSC), and relative strand cross-correlation coefficient (RSC) [17]. After removing PCR duplicates and unaligned reads, uniquely mapped reads of each ChIP sample were used to call peaks against the input sample using MACS2 [20] with default parameters (*p*-value < 0.02). After peak calling, the irreproducibility discovery rate (IDR) (https://github.com/nboley/idr, accessed on 6 August 2016) was performed to identify the reproducible ChIP-seq peaks, which have high consistency between two biological replicates (*p*-value < 0.02). ChIP-seq tracks were visualized in IGV [21] using the bigWig file that includes ChIP-seq signals and IDR-passed ChIP-seq peak file.

The genomic distribution analysis was done by overlapping the IDR-passed reproducible MYBL2 or FOXM1 ChIP-seq peaks with the promoter regions of known genes (defined as 2 kb windows centered on transcription start sites of the canonical transcripts), histone marks H3K4me3 and H3K27ac, and CTCF ChIP-seq peaks downloaded from ENCODE (Appendix A). The canonical transcript information was obtained from biomaRt [22] using GENCODE V29 reference genome annotation [23]. To determine the genomic distribution of MYBL2 or FOXM1 binding sites, the IDR-passed reproducible MYBL2 or FOXM1 ChIP-seq peaks, which were within 2 kb windows of transcription start sites or overlapped with H3K4me3 ChIP-seq peaks, were categorized as promoter peaks. The non-promoter peaks that overlapped with H3K27ac ChIP-seq peaks were categorized as enhancer peaks, and the rest of the non-promoter peaks that overlapped with CTCF ChIP-seq peaks were categorized as insulator peaks. The peaks which were not overlapped with the above features were categorized as other peaks.

The common and unique MYBL2 and FOXM1 ChIP-seq peaks were identified using BEDTools’ [24] intersectBed function. The actual ChIP-seq signals on the identified common and unique peak regions were visualized via deepTools [25]. Motif analysis was performed using the findMotifsGenome.pl function in Homer [26] with the default background sequence to determine the enriched motifs (*p*-value ≤ 0.01) in all common and unique IDR-passed MYBL2 and FOXM1 peaks. (Appendix A). The most enriched motifs of each category were determined by ranking with *p*-values and listed in Appendix A.

### 2.7. siRNA Transfection

MYBL2 and FOXM1 were knocked down using 25 nM of small interfering RNAs (siRNAs). Negative control scrambled siRNA (Cat #D-001810-10-05) and siRNAs targeting MYBL2 (Cat #L-010444-00-0005) or FOXM1 (Cat #L-009762-00-0005) were purchased from Dharmacon (Horizon Discovery, Waterbeach, UK). A549 lung adenocarcinoma cells were transfected with control scrambled siRNA or siRNAs targeting MYBL2 or FOXM1 or a co-transfection of both using DharmaFECT1 transfection reagent (Cat #T-2001-02, Horizon Discovery, Waterbeach, UK) according to the manufacturer’s instructions as we previously performed siRNA experiments [15]. RNA was isolated using the Aurum total RNA mini kit (Cat #7326820, Bio-Rad, Hercules, CA, USA), following the manufacturer’s manual.

### 2.8. RNA-Seq

RNA was isolated from cells using TRIzol reagent (Cat #R2053, Zymo Research, Irvine, CA, USA), and the quality of RNA was assessed by calculating RIN (RNA integrity number) using a 2100 Bioanalyzer system (Agilent Technologies, Santa Clara, CA, USA). QC-passed RNA was used to make RNA-seq libraries, and sequencing was performed using an Illumina NovaSeq 6000. At least three biological replicates of each experiment were prepared and analyzed. RNA-seq reads were processed using an in-house mRNA-seq analysis pipeline, which was adapted from the NCI Genomic Data Commons bioinformatic pipeline (https://gdc.cancer.gov/, accessed on 11 April 2021). Reads were trimmed using TrimGalore (https://github.com/FelixKrueger/TrimGalore, accessed on 11 April 2021) to filter out low-quality reads and the adapter sequences. The trimmed reads were then aligned to human reference genome version 38 (hg38) with the annotation GENCODE V29 [23] using STAR [27] to generate bam files. HTseq [28] was used to quantify the read counts for each gene, with the bam files as the input. DEseq2 [29] is used to calculate the significance of gene expression difference for each gene between experimental and control groups. The *p*-values were then adjusted to correct multiple comparisons and measure the false discovery rate (FDR). The differentially expressed genes (DEGs) were identified based on FDR and fold change (FDR < 0.05, |Fold Change| > 1.5) (Appendix A). Genes directly regulated by MYBL2 or FOXM1 were identified by overlapping genes that are downregulated by siMYBL2 or siFOXM1 with the gene promoters that are bound by MYBL2 or FOXM1. Promoters were defined as 2 kb windows of the transcription start sites as described above. Ingenuity pathway analysis (IPA) [30] was performed to determine the signaling pathways, and the enriched signaling pathways were identified using FDR and listed in Appendix A.

We obtained fastq files of RNA-seq data of IMR-90 and AG04450 from the ENCODE and alveolar epithelial cell lines (AEC) from the GEO (Appendix A). These were processed to quantify gene expression levels with the above-described RNA-seq analysis pipeline. *MYBL2* gene expression levels were plotted using log2 transformed TPM values calculated by Rsem [31]. The TCGA lung adenocarcinoma (LUAD) gene expression data [32] were used to measure the expression levels of the identified MYBL2 and FOXM1 target genes in normal lung tissue and lung adenocarcinoma tissue samples. TCGA RNA-seq data were obtained from the NCI Genomic Data Commons Data portal (https://portal.gdc.cancer.gov, accessed on 24 September 2020) and visualized in a heatmap using log2-transformed HTseq read count data. Student’s *t*-test, the Wilcoxon rank sum test, and the Wald test were performed to measure the statistical significance of the difference between normal and tumor tissue samples.

### 2.9. RT-qPCR

Total RNA was isolated from A549 and NCI-H2126 cells using TRIzol reagent (Cat #R2053, Zymo Research, Irvine, CA, USA), and 1 μg of it was converted to cDNA using the iScript Reverse Transcription Supermix (Cat# 1708841, Bio-Rad, Hercules, CA, USA) according to the manufacturer’s instructions. Quantitative real-time PCR was performed on a Bio-Rad CFX96 Real-Time PCR Detection System. The primer sequences are listed in Appendix A. The data were analyzed by the 2^(−^^ΔΔCt)^ method and represented as fold changes of gene expression relative to *ACTB*.

### 2.10. Western Blotting

Cells were lysed with lysis buffer (RIPA lysis buffer) containing a protease inhibitor mixture (Cat #4693192002, MilliporeSigma, Carlsbad, CA, USA). Lysates were collected after centrifugation at 12,000 rpm for 15 min at 4 °C. For nuclear-fraction extraction, cells were lysed with cytoplasmic extraction buffer and then centrifuged at 430 rcf for 5 min. The supernatant was removed, and the nuclear pellets were further lysed with 1X RIPA buffer. Protein concentrations were determined using the Qubit (Cat #Q33212, ThermoFisher Scientific, Waltham, MA, USA). Protein extracts (40 μg) were separated in 4–15% SDS-polyacrylamide gels (Cat #4561085, Bio-Rad, Hercules, CA, USA) and then transferred onto nitrocellulose membranes (GE Healthcare Life Sciences, Buckinghamshire, UK). After blocking for 60 min in 5% skim milk, the membranes were incubated with the appropriate primary antibodies and fluorescent-conjugated secondary antibodies. Western blots were visualized using the Odyssey CLx Infrared imaging system (LI-COR Biosciences, Lincoln, NE, USA). Signal intensity was measured using ImageJ software (National Institutes of Health, Bethesda, MD, USA) and normalized against ACTB signals. Statistical significance was determined using Student’s *t*-test. Results are presented as mean ± SE, and a *p*-value cut-off of 0.05 was considered statistically significant.

### 2.11. Survival Analysis of Lung Adenocarcinoma Patients

For survival analysis, correlation analyses between the expression of MYBL2, FOXM1, and CENPA and patient survival data were performed using TCGA LUAD tissue samples (*n* = 535). Gene expression data (HTseq count) and corresponding patient clinical survival data were downloaded from the NCI Genomic Data Commons Data portal (https://portal.gdc.cancer.gov, accessed on 24 September 2020). The HTseq count data was normalized by log2 transformation, and the patients with expression of each gene in the highest or lowest quartiles were grouped. For each gene, a survival analysis was performed between the two groups using the Kaplan–Meier survival curve and the log-rank test from the survival R package (https://CRAN.R-project.org/package=survival, accessed on 18 April 2022).

### 2.12. Statistical Analysis

Statistical analyses for Western blot and RT-qPCR experiments were performed using the Student’s *t*-test. The differential expression analysis of the *MYBL2* gene between A549 lung adenocarcinoma cells and other lung non-cancer lung cells was performed using the Student’s *t*-test, and *p*-values were adjusted using the Holm–Bonferroni method. The expression difference among all expressed genes, genes bound, and genes not bound by MYBL2 was measured by using the Student’s *t*-test. Differential gene expression analysis for RNA-seq datasets was performed using the Wald test, and *p*-values were adjusted using the Benjamini–Hochberg method. The expression difference of the target genes of MYBL2 or FOXM1 between LUAD and normal tissues was measured by using Student’s *t*-test, the Wilcoxon rank sum test, and the Wald test. The survival analysis was performed using the log-rank method.

## 3. Results

### 3.1. MYBL2 Binds to the Promoters of Highly Expressed Genes in Lung Adenocarcinoma Cells

Our previous study that integrated RNA-seq and DNA methylation datasets from lung adenocarcinoma tissues suggested that MYBL2 is a key transcriptional regulator linked to lung cancer [7]. Moreover, overexpression of MYBL2 is associated with poor survival of lung adenocarcinoma patients [7] (Appendix A). To study the molecular mechanisms of MYBL2 in lung adenocarcinoma cells, we first investigated MYBL2 expression levels in normal alveolar epithelial cells (AEC), airway epithelial cells, lung fibroblast cells (IMR-90, AG04450), and lung adenocarcinoma cells (A549). We found that A549 cells exhibited higher MYBL2 expression compared to non-cancer cells (Figure 1A) (*p*-value < 0.05). MYBL2 is a transcription factor that includes the Myb-type HTH DNA-binding domain. Therefore, we performed ChIP-seq using MYBL2 antibodies in A549 cells. After checking the quality of ChIP-seq datasets following the ENCODE ChIP-seq guidelines [17], we identified two biological replicates that passed QC (Appendix A) and identified a total of 482 reproducible, robust binding sites of MYBL2 (Figure 1B, Appendix A). To determine the genomic distribution of MYBL2 binding sites in A549 cells, we compared MYBL2 ChIP-seq peaks with H3K4me3 (promoter mark), H3K27ac (enhancer mark), and CTCF (insulator mark) ChIP-seq peaks, which were generated in the same A549 cells (Appendix A). The genomic distribution analysis of MYBL2 binding sites showed that most binding sites (85.68%) are located at promoters (Figure 1B). Further investigation of gene expression levels revealed that genes whose promoters are bound by MYBL2 (MYBL2-bound) are more highly expressed than genes that are not bound by MYBL2 at their promoters (non-bound) or all expressed genes, on average (Figure 1C) (*p*-value < 0.0001). By integrating with TCGA lung adenocarcinoma RNA-seq data, we also confirmed that genes whose promoters bound by MYBL2 are expressed higher than those not bound by MYBL2 in lung adenocarcinoma cells (Appendix A). This suggests that MYBL2 is a crucial transcription factor that regulates oncogenes in lung adenocarcinoma.

Next, to identify genes that are regulated by MYBL2, we performed siRNA knockdown experiments followed by RNA-seq, generating 4 siMYBL2 RNA-seq replicates and 4 siControl RNA-seq replicates (Figure 1D). We identified 1439 differentially expressed genes (DEGs) that include 629 downregulated and 810 upregulated genes by knockdown of MYBL2 (fold change > 1.5, false discovery rate (FDR) < 0.05) (Figure 1D, Appendix A). We further analyzed RNA-seq data integrating with MYBL2 ChIP-seq data and found that more genes that are downregulated by MYBL2 depletion are bound by MYBL2 at their promoters than genes upregulated by MYBL2 depletion, indicating that MYBL2 acts as a transcriptional activator (Figure 1E). From the list of the identified genes regulated by MYBL2, we found FOXM1 (Figure 1F, Appendix A). Since overexpression of FOXM1 is associated with poor patient survival in lung adenocarcinoma [7] (Appendix A), we selected FOXM1 for further examination.

### 3.2. FOXM1 Binds to the Active Promoters Bound by MYBL2 in Lung Adenocarcinoma Cells

FOXM1 is a transcription factor that belongs to the forkhead transcription factor family, and it is one of the key transcriptional regulators linked to lung adenocarcinoma [7]. To characterize the molecular mechanisms of FOXM1 in lung adenocarcinoma cells, we profiled the genome-wide binding sites of FOXM1 in A549 cells using ChIP-seq. After checking the quality of ChIP-seq datasets according to the ENCODE ChIP-seq guidelines [17], we identified a total of 254 reproducible, robust binding sites of FOXM1 in A549 cells (Figure 2A, Appendix A). We found that 76.8% of the FOXM1 binding sites are located within promoters, while 21.6% of them are in distal regions, which are not enhancers or insulators (Figure 2B). Overlap analysis between FOXM1 and MYBL2 ChIP-seq revealed that 66% of the FOXM1 binding sites are shared with MYBL2 binding sites (Figure 2C top). We found that FOXM1 and MYBL2 ChIP-seq signals at both common and unique binding sites are robust and reproducible (Figure 2C bottom). However, the average ChIP-seq signals at the binding sites that are shared between FOXM1 and MYBL2 (common) are stronger than unique binding sites (Figure 2D,E). The most enriched motifs at commonly bound sites were the cell cycle gene homology region (CHR) and the CCAAT motif (common promoter elements) (Figure 2F). On the other hand, we identified that the TLX nuclear receptor motif was more enriched at FOXM1 unique binding sites, while MYBL2 (B-MYB), E2F, and ETS motifs are more enriched at MYBL2 unique binding sites. The results of the ChIP-seq and transcription factor motif enrichment analysis indicate that MYBL2 and FOXM1 bind to promoters of cell cycle genes together in lung adenocarcinoma while each factor may bind to unique sites, possibly working with distinct transcription factor partners for the other roles.

### 3.3. MYBL2 and FOXM1 Regulate Cell Cycle Genes in Lung Adenocarcinoma Cells

To identify genes regulated by FOXM1, we performed siRNA experiments to knock down FOXM1, and then RNA-seq was conducted (Figure 3A). After generating and analyzing 4 siFOXM1 RNA-seq replicates and 4 siControl RNA-seq replicates, we found that 162 genes are differentially expressed (133 downregulated and 29 upregulated genes) upon knockdown of FOXM1 (fold change > 1.5, FDR < 0.05) (Appendix A). Integrative analysis of ChIP-seq and RNA-seq upon FOXM1 knockdown identified that more downregulated genes are bound by FOXM1 compared to upregulated genes, suggesting that FOXM1 also acts as a transcriptional activator (Appendix A). Next, we further examined how the knockdown of both MYBL2 and FOXM1 (siMYBL2 and siFOXM1) affects the gene expression levels (Figure 3B and Appendix A). Compared to the individual knockdown of FOXM1, more genes were significantly changed upon knockdown of FOXM1 and MYBL2 together (siMYBL2 and siFOXM1) (*n* = 1191) (664 downregulated genes and 527 upregulated genes, fold change > 1.5, FDR < 0.05) (Figure 3B). This additive effect upon knockdown of both FOXM1 and MYBL2 suggests that these factors work together as transcriptional activators in lung adenocarcinoma cells.

Genes whose promoters are commonly bound by MYBL2 and FOXM1 were enriched with CHR (cell cycle gene homology region) motifs. Signaling pathway analysis of the genes dysregulated upon knockdown of MYBL2, FOXM1, or both MYBL2 and FOXM1 in lung adenocarcinoma revealed that the genes regulated by MYBL2 and FOXM1 were mainly involved in cell cycle processes (cell survival, DNA replication, and DNA repair), cellular assembly and organization, and cellular development (Figure 3C–E). We also found that more genes involved in cell cycle-related pathways are affected by siMYBL2 and siFOXM1 (Figure 3E). Furthermore, we investigated which cell-cycle phases are controlled most by MYBL2 and FOXM1. Among cell-cycle phases, we found that genes involved in the G2/M phase were more enriched in FOXM1 DEGs, while genes involved in the G1/S phases were more enriched in MYBL2 DEGs (Figure 3F).

### 3.4. Identification of MYBL2 and FOXM1 Target Genes in Lung Adenocarcinoma Cells

To further identify genes directly regulated by MYBL2 and FOXM1, we integrated RNA-seq data of siMYBL2, siFOXM1, and siMYBL2 and siFOXM1 with MYBL2 and FOXM1 ChIP-seq data. As both factors have properties of transcriptional activators, we narrowed down the list of genes whose promoters are bound by MYBL2 or FOXM1 and whose expression levels were decreased upon knockdown of MYBL2 or FOXM1 (Figure 4A). From this analysis, we identified that 108 genes are directly activated by MYBL2, FOXM1, or both (Figure 4A, Appendix A). Among the identified 108 genes, we found 10 genes (*CENPA*, *FAM83D*, *KNSTRN*, *NEURL1B*, *CCNF*, *CDKN3*, *CCNB1*, *BORA*, *CDK1*, and *CCNA2*) whose expression levels went down in both single and double knockdown experiments (Figure 4B).

To validate the identified common target genes, an siRNA-mediated knockdown RT-qPCR experiment was newly performed in A549 and NCI-H2126 adenocarcinoma cells (Figure 4C,D). Most of the genes were significantly reduced by the knockdown of both MYBL2 and FOXM1 in A549 and NCI-H2126 lung adenocarcinoma cells. We further examined the expression levels of the identified MYBL2 and FOXM1 target genes in lung adenocarcinoma tissues and adjacent normal lung tissues using TCGA RNA-seq datasets (Figure 4E). We revealed that all of the 10 target genes were highly expressed in lung tumor tissues from lung adenocarcinoma patients compared to normal tissues. We further examined if the expression level of the identified target genes is associated with survival rates (Appendix A). We found that overexpression of these 8 genes (*CENPA, FAM83D, KNSTRN, CDKN3, CCNB1, CDK1*, and *CCNA2*) is significantly associated with poor survival of lung adenocarcinoma patients (*p*-value < 0.05). For example, overexpression of CENPA, which is identified as one of the key transcriptional regulators for lung adenocarcinoma, is associated with poor survival of lung adenocarcinoma patients [7] (Figure 4F). When we further examined the protein expression of CENPA using Western blot in A549 and NCI-H2126 lung adenocarcinoma cells (Figure 4G), we found that the CENPA protein level was decreased upon siMYBL2 and siFOXM1 treatment. Based on these results, it is suggested that both MYBL2 and FOXM1 transcription factors cooperate to stimulate the expression of CENPA and other cell-cycle genes.

### 3.5. FDI-6 Alters Cell Proliferation and Inhibits the Activities of FOXM1 and MYBL2 in Lung Adenocarcinoma Cells

Next, we assessed potential treatment options for lung adenocarcinoma targeting MYBL2 and FOXM1. We found that a small compound called FDI-6 (forkhead domain inhibitor 6) is reported to inhibit the binding of FOXM1 to DNA in breast cancer cells [33]. However, the effect of FDI-6 in lung adenocarcinoma is not yet characterized. Therefore, we tested the effect of FDI-6 in A549 and NCI-H2126 lung adenocarcinoma cells. When we investigated cell growth rate by treating lung adenocarcinoma cells with various concentrations of FDI-6, we found that FDI-6 effectively inhibits cell growth in a dose-dependent manner (Figure 5A). For example, the growth of A549 cells was inhibited upon treatment with 20 μM FDI-6 at 48 h and 72 h. Next, we examined the long-term anti-proliferative effect of FDI-6 by performing a colony formation assay (Figure 5B and Appendix A). We found that colony formation in both A549 and NCI-2126 cells treated with FDI-6 is diminished. Remarkably, when cells were treated with 20 μM of FDI-6, colonies were almost completely gone. Collectively, we found that FDI-6 suppressed lung adenocarcinoma cell growth as well as the clonogenic potential of lung adenocarcinoma cells in a dose-dependent manner.

Furthermore, we tested whether FDI-6 affects FOXM1 activity in A549 lung adenocarcinoma cells by performing Western blot experiments using FDI-6-treated nuclear extracts. The protein level of FOXM1 within the nucleus was reduced in FDI-6-treated nuclear extracts compared to the control (Figure 5C and Appendix A, *p*-value ≤ 0.01). Interestingly, we also found that MYBL2 protein level in nuclear extracts was also reduced upon FDI-6 treatment in A549 lung adenocarcinoma cells. This indicates that treatment with FDI-6 can reduce the activities of FOXM1 as well as MYBL2 in nuclei of the lung adenocarcinoma cells.

Lastly, to determine the effect of FDI-6 on the lung adenocarcinoma transcriptome, RNA-seq was performed after FDI-6 treatment. We identified that 1827 DEGs, which include 739 downregulated and 1088 upregulated genes, are changed upon treatment (Figure 5D, Appendix A). Signaling pathway analysis of genes changed upon FDI-6 treatment showed that gene sets involved in cell cycle, cell death and survival are differentially expressed, which has a high similarity to siRNA signaling pathway analysis results (Figure 5E). We further examined which cell-cycle phases are controlled most by FDI-6 treatment. The genes involved in the G2/M phase were more enriched in DEGs upon FDI-6 treatment (Figure 5F).

Furthermore, we investigated if the expression levels of MYBL2 and FOXM1 target genes were changed by FDI-6 treatment. Among MYBL2 and FOXM1 target genes, we found that the expression levels of *CENPA* and *NEUR1LB* are greatly suppressed by FDI-6 treatment (Appendix A). In conclusion, it is suggested that FDI-6 can change the expression of genes involved in cell cycle regulation, affecting the activity of both FOXM1 and MYBL2 in lung adenocarcinoma cells.

## 4. Discussion

MYBL2 is a transcription factor that can have either pro-survival or anti-survival functions in a cell-type-specific manner. MYBL2 is reported to be associated with the prognosis of lung adenocarcinoma [7,34] (Appendix A). However, it is not clear how the MYBL2 transcription factor regulates genes and leads to carcinogenesis in lung adenocarcinoma. In this study, we mapped the global binding sites of MYBL2 and found that MYBL2 binds to the promoters of genes that are highly expressed in lung adenocarcinoma (Figure 1). Moreover, by using MYBL2 ChIP-seq and knockdown experiments coupled with RNA-seq, we showed that MYBL2 acts as a transcriptional activator in lung adenocarcinoma cells (Figure 1). Furthermore, we showed that the binding sites of another transcriptional activator, FOXM1, are largely shared with MYBL2 binding sites in lung adenocarcinoma cells (Figure 2). The most enriched motifs of the MYBL2 and FOXM1 commonly bound sites were CCAAT and CHR motifs (Figure 2). Moreover, knocking down MYBL2 and FOXM1 together resulted in more changes in genes involved in the cell cycle than each individual knockdown (Figure 3). These results indicate that FOXM1 and MYBL2 work together to regulate cell-cycle-related genes, potentially interacting with each other in lung adenocarcinoma cells.

In accordance with the previous finding that MYBL2 and FOXM1 control genes expressed in the G2/M phase of the cell cycle [35,36], we found that MYBL2 and FOXM1 regulate genes belonging to the G2/M phase in lung adenocarcinoma cells (Figure 3). For example, we showed that *CENPA* is one of the key targets of MYBL2 and FOXM1 in lung adenocarcinoma cells (Figure 4). CENPA, which is a centromeric histone H3 variant, is a key cell cycle-regulated gene. It is transcribed during the G2/M phase in humans [37], and it is reported that DNA replication, centromere breakage, recombination, and chromosome translocations are delayed in the absence of CENPA [38]. CENPA is also known to be recruited to double-strand DNA breaks [39]. *CENPA* overexpression is often found to be in different types of cancer [40], but its mechanism is largely unknown. We have shown that both gene and protein expression levels of CENPA were effectively reduced in the siRNA-mediated knockdown of MYBL2 and FOXM1 (Figure 4). These results suggest that MYBL2 and FOXM1 work together to regulate CENPA.

In addition to CENPA, we identified additional potential target genes of MYBL2 and FOXM1 in lung adenocarcinoma (Figure 4). For example, neuralized E3 ubiquitin protein ligase 1B (NEURL1B) is known to act as a tumor suppressor gene in colon cancer and medulloblastoma [41,42]. However, in our study, we found that *NEURL1B* acts more like an oncogene in lung adenocarcinoma cells. This is consistent with the previous study (Figure 4), which reported it as a putative brain metastasis-promoting gene in lung cancer cells [43]. We also found that cyclin B1 (CCNB1) is regulated by MYBL2 and FOXM1 (Figure 4), and overexpression of *CCNB1* is associated with poor survival of lung adenocarcinoma patients (Appendix A). *CCNB1* was reported as one of the driver genes for lung cancer from the bioinformatic meta-analysis of gene expression data of non-small cell lung cancer with protein-protein interaction data [35]. CCNB1 interacts with CDK1 to form a complex that phosphorylates their substrates and promotes the G2/M transition in the cell cycle. Inhibiting the expression of CCNB1 promotes apoptosis in colorectal cancer cells [44]. Newly identified genes controlled by MYBL2 and FOXM1 using ChIP-seq and RNA-seq will help elucidate the molecular mechanisms of lung carcinogenesis. Moreover, further characterization of how these genes are connected will provide important insights into carcinogenesis.

To target the oncogenic activities of MYBL2 and FOXM1 in lung adenocarcinoma, we have explored ways to suppress them using an inhibitor. FDI-6 has been identified as a FOXM1-specific inhibitor in breast cancer cells by blocking the DNA binding activity of FOXM1, and its effect is selective for FOXM1 target genes without affecting genes regulated by homologous forkhead family factors [33]. FOXM1 protein levels within the nucleus were decreased by FDI-6 treatment, resulting in the inhibition of cell proliferation, migration, and invasion [33]. Several studies have examined the anti-cancer effect of FDI-6 on breast and ovarian cancer cells [33,45,46]. However, its anti-cancer effect on lung adenocarcinoma cells has not yet been investigated. We have shown that FDI-6 treatment effectively inhibited cell proliferation in A549 and NCI-H2126 lung adenocarcinoma cells. A549 lung adenocarcinoma cells treated with FDI-6 show significant reductions in FOXM1 protein accumulation in nuclei (Figure 5). Additionally, we observed that FDI-6 could suppress the MYBL2 protein level in A549 lung adenocarcinoma nuclei (Figure 5). This may be due to the fact that FDI-6 treatment inhibits the binding of both MYBL2 and FOXM1 to DNA as they interact with each other and share their binding sites. Further studies are needed to explain this decrease and determine whether FDI-6 effectively targets lung cancer cells. However, our results, which show that FDI-6 treatment changes the expression of key genes and proteins involved in the cell cycle, suggest that MYBL2 and FOXM1 are potential treatment targets for lung adenocarcinoma.

## 5. Conclusions

Previous studies have reported that MYBL2 and FOXM1 may play an important role in carcinogenesis [8,47,48,49,50]. However, their targets or molecular mechanisms in lung adenocarcinoma have not yet been well-characterized. In this study, we performed MYBL2 and FOXM1 ChIP-seq, identifying global binding sites of MYBL2 and FOXM1 in lung adenocarcinoma cells for the first time. By integrating the ChIP-seq data with RNA-seq data generated upon the knockdown of MYBL2 and FOXM1, we revealed that MYBL2 and FOXM1 act as transcriptional activators in lung adenocarcinoma. Moreover, we identified novel downstream targets of MYBL2 and FOXM1 that include key oncogenes such as *CENPA* in lung adenocarcinoma. We also provided insights into how MYBL2, FOXM1, and cell-cycle genes, which are critical for lung cancer patient survival, are regulated in lung adenocarcinoma cells. Our results demonstrated that MYBL2 and FOXM1 cooperatively regulate the G2/M phase transition cell-cycle genes in lung adenocarcinoma cells. Furthermore, we showed that FDI-6 alters cell proliferation and inhibits the activities of FOXM1 as well as MYBL2 in lung adenocarcinoma cells. Further studies to characterize the mechanisms of MYBL2 and FOXM1 target genes and to determine the effectiveness of FDI-6 in lung adenocarcinoma patients are needed. Collectively, findings from this study will help the development of new therapeutic strategies for lung cancer.

## Figures and Tables

**Figure 1 cancers-14-04979-f001:**
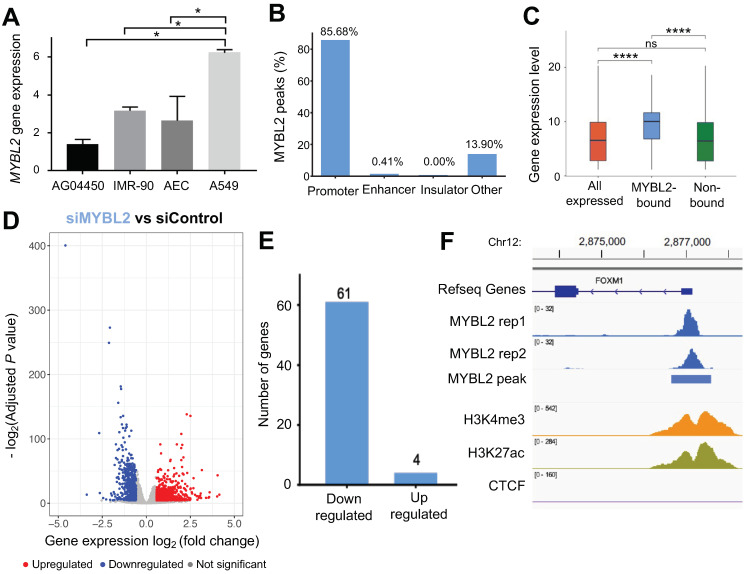
MYBL2 binds to the promoters of highly expressed genes in lung adenocarcinoma cells. (**A**) A barplot shows the expression level of *MYBL2* in AG04450, IMR-90, AEC and A549 cells. The *p*-value is calculated using Student’s *t*-test, and the *p*-value is adjusted by using the Holm–Bonferroni method (*, *p*-value < 0.05). (**B**) Genomic distribution of MYBL2 binding sites in A549 cells. (**C**) A boxplot showing the expression level of all genes expressed in A549 cells, genes with MYBL2-bound promoters, and genes without MYBL2-bound promoters (****, *p*-value < 0.0001). (**D**) A volcano plot demonstrating differential gene expression after knockdown of MYBL2 in A549 cells. *X*-axis shows the log2 fold change between MYBL2 siRNA (siMYBL2) and control scrambled siRNA (siControl) cells, and *Y*-axis shows the −log2(adjusted *p*-value) upon the comparison between siMYBL2 and siControl. The red dots represent the upregulated genes, the blue dots represent the downregulated genes, and the gray dots represent the genes that are not differentially expressed (−log2(adjusted *p*-value) > 0.05 and fold change < 1.5). (**E**) A barplot shows the number of downregulated genes (**left**) and upregulated genes (**right**) with MYBL2-bound promoters. (**F**) A genome browser screenshot near the *FOXM1* gene, showing two replicates of MYBL2 ChIP-seq signals and IDR passed MYBL2 peak, H3K27ac, H3K4me3, and CTCF ChIP seq signals in A549 cells.

**Figure 2 cancers-14-04979-f002:**
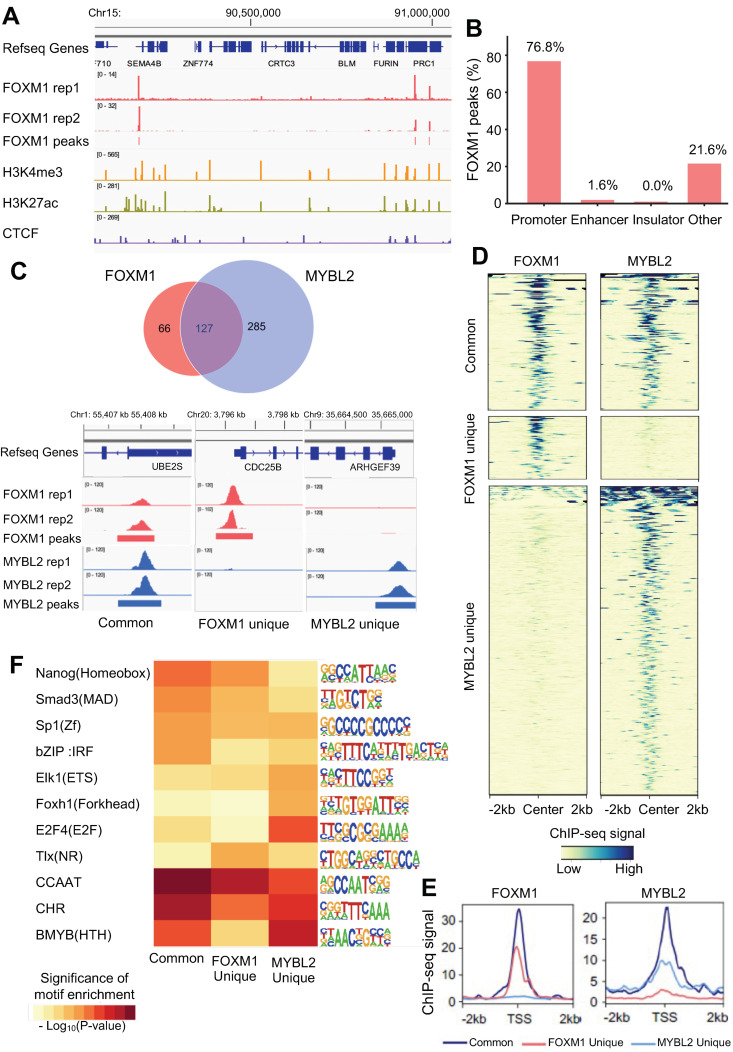
FOXM1 binds to the active promoters bound by MYBL2 in lung adenocarcinoma cells. (**A**) A genome browser screenshot of the chr15q26 region showing two replicates of FOXM1 ChIP-seq signals, IDR-passed FOXM1 peaks, H3K27ac, H3K4me3, and CTCF ChIP-seq signals in A549 cells. (**B**) Genomic distribution of FOXM1 binding sites in A549 cells. (**C**) A Venn diagram of the overlapped binding sites between MYBL2 and FOXM1 (**top**). Example genome browser screenshots of the common (**bottom left**), FOXM1-unique (**bottom middle**), and MYBL2-unique (**bottom right**) binding sites. (**D**) A heatmap showing FOXM1 (**left**) and MYBL2 (**right**) ChIP-seq signals within a 2 kb window of the center of common (**top**), FOXM1-unique (**middle**), and MYBL2-unique (**bottom**) binding sites. The ChIP-seq signals are indicated by the color from yellow (**low**) to blue (**high**). (**E**) The average ChIP-seq signals of FOXM1 (**left**) and MYBL2 (**right**) at common (dark blue), FOXM1-unique (red), and MYBL2-unique (light blue) binding sites located within a 2 kb window of the transcription start sites (TSS). (**F**) A heatmap showing the most enriched motifs found in either common (**left**), FOXM1-unique (**middle**) or MYBL2-unique (**right**) binding sites. The motif analysis is performed using the 100 bp window around the ChIP-seq peak summits. Shown to the right are the sequence logos of the motifs. The statistical significance of each motif enrichment of motifs is indicated by a color gradient from yellow (**low**) to dark red (**high**).

**Figure 3 cancers-14-04979-f003:**
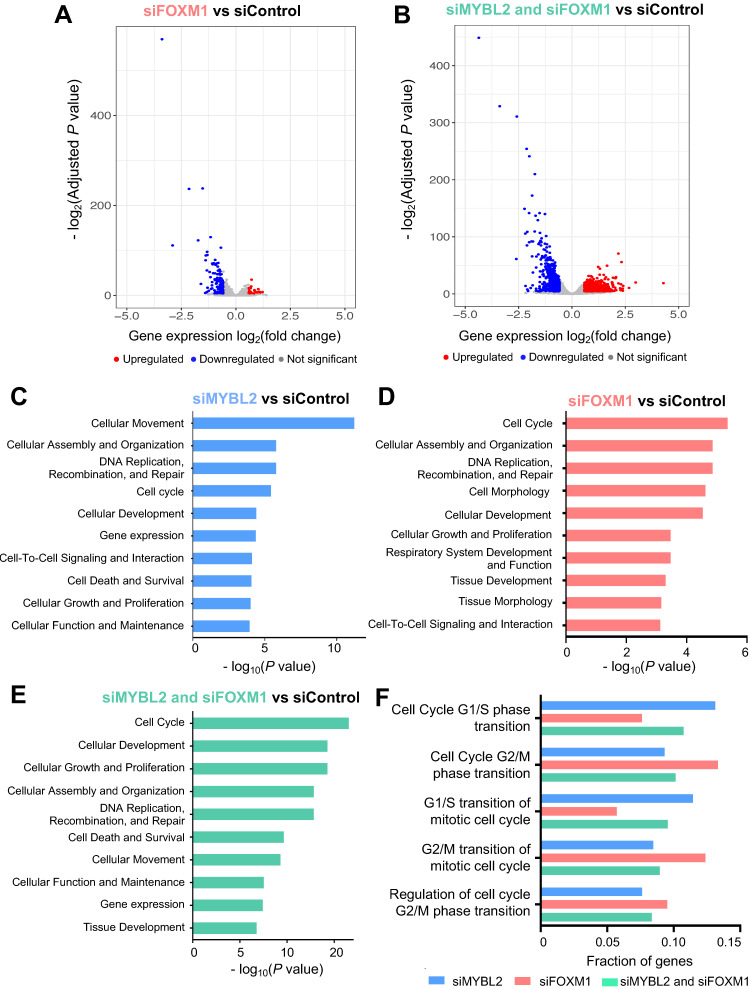
FOXM1 activates cell cycle-related genes together with MYBL2 in lung adenocarcinoma cells. (**A**) A volcano plot demonstrating differential gene expression after knockdown of FOXM1 in A549 cells. *X*-axis shows the log2 fold change between FOXM1 siRNA (siFOXM1) and control scrambled siRNA (siControl) cells, and *Y*-axis shows the −log2(adjusted *p*-value) upon the comparison between siFOXM1 and siControl. The red dots represent the upregulated genes, the blue dots represent the downregulated genes, and the gray dots represent the genes that are not statistically significantly differentially expressed (−log2(adjusted *p*-value) > 0.05 and fold change < 1.5). (**B**) A volcano plot demonstrating differential gene expression after knockdown of both MYBL2 and FOXM1 (siMYBL2 and siFOXM1) in A549 cells. *X*-axis shows the log2 fold change between siMYBL2 and siFOXM1 and siControl cells, and *Y*-axis shows the −log2(adjusted *p*-value) upon the comparison between siMYBL2 and siFOXM1 and siControl. The red dots represent the upregulated genes, the blue dots represent the downregulated genes, and the gray dots represent the genes that are not statistically significantly differentially expressed (−log2(adjusted *p*-value) > 0.05 and fold change < 1.5). (**C**–**E**) Barplots showing the top 10 biological functional categories, in which genes changed by siMYBL2, siFOXM1, and siMYBL2 and siFOXM1 are most enriched. (**F**) Fraction of genes whose expression levels are changed upon siMYBL2, siFOXM1, and siMYBL2 and siFOXM1 and involved in each cell cycle phase are shown.

**Figure 4 cancers-14-04979-f004:**
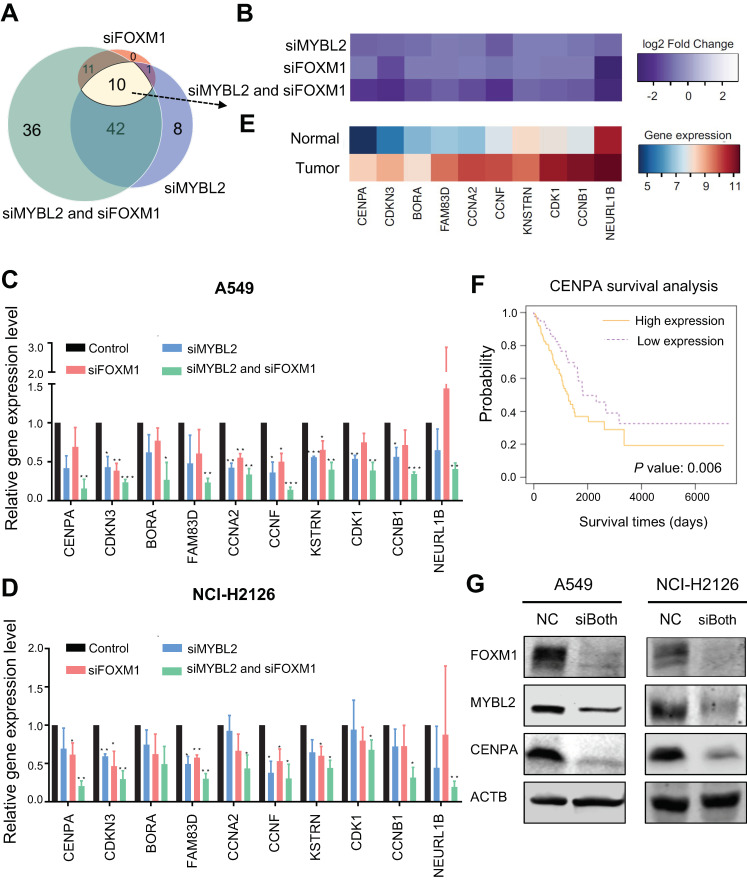
Identification of MYBL2 and FOXM1 target genes in lung adenocarcinoma cells. (**A**) A Venn diagram showing the overlapped genes among the MYBL2 and FOXM1 target genes, identified by the downregulated genes in siFOXM1 (pink), siMYBL2 (blue), and siMYBL2 and siFOXM1 (green) experiments with promoters bound by MYBL2, FOXM1, or both. (**B**) A heatmap showing the log2 fold change gene expression of the identified MYBL2 and FOXM1 target genes from the knockdown RNA-seq experiments. siFOXM1, siMYBL2, and siMYBL2 and siFOXM1 RNA-seq data are compared to corresponding siControl RNA-seq data. Validation of downstream targets of both MYBL2 and FOXM1 using RT-qPCR upon knockdown experiments in (**C**) A549 cells and (**D**) NCI-H2126 cells. Data represent the mean SE (*n* = 3). *, *p*-value ≤ 0.05, **, *p*-value ≤ 0.01, ***, *p*-value ≤ 0.001. (**E**) A heatmap showing the expression levels of the identified MYBL2 and FOXM1 target genes in TCGA LUAD RNA-seq datasets. (**F**) A survival analysis indicates that overexpression of CENPA is associated with poor survival of lung adenocarcinoma patients (*p*-value = 0.006). TCGA LUAD RNA-seq and clinical data were used to perform this analysis. (**G**) FOXM1, MYBL2, and CENPA protein expression levels were analyzed upon knockdown of MYBL2 and FOXM1 by Western blot (**right**) (*n* = 3). ACTB is used as a control. NC: negative control.

**Figure 5 cancers-14-04979-f005:**
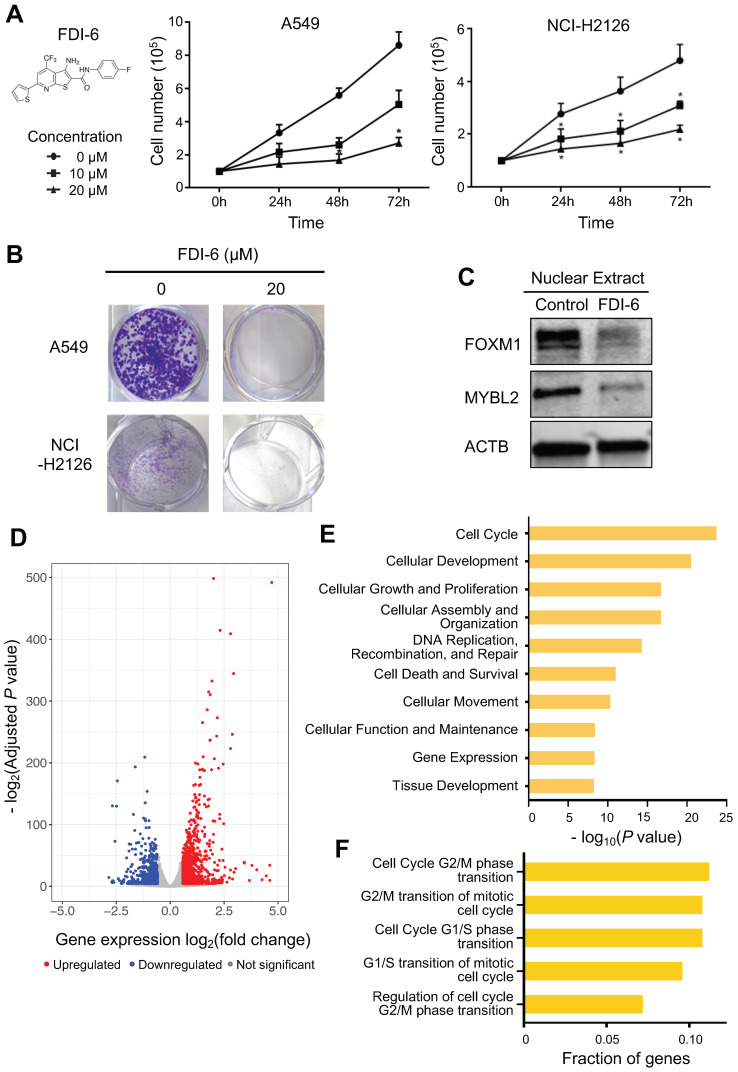
FDI-6 alters cell proliferation and inhibits the activities of FOXM1 and MYBL2 in lung adenocarcinoma cells. (**A**) A structure of FDI-6 (**left**). A549 and NCI-H2126 cells were counted every 24 h after treatment with different concentrations of FDI-6 for 72 h (*n* = 3). Data represent the mean SE (*n* = 3). Statistical significance was determined using Student’s *t*-test. *, *p*-value ≤ 0.05 vs. control. (**B**) Colony formation images of A549 and NCI-H2126. A549 and NCI-H2126 cells after being treated with FDI-6 for 24 h and then incubated for 14 days. (**C**) FOXM1 and MYBL2 protein expression levels were analyzed in the nuclear extract of A549 cells by Western blot (**right**) (*n* = 3). ACTB is used as a control. (**D**) A volcano plot demonstrating differential gene expression after treatment of FDI-6 in A549 cells. *X*-axis shows the log2 fold change between FDI-6 treated and untreated cells, and *Y*-axis shows the −log2(adjusted *p*-value) upon the comparison between FDI-6 treated and untreated cells. The red dots represent the upregulated genes, the blue dots represent the downregulated genes, and the gray dots represent the genes that are not statistically significantly differentially expressed (−log2(adjusted *p*-value) > 0.05 and fold change < 1.5). (**E**) A barplot showing the top 10 biological functional categories in which FDI-6 target genes (identified by downregulated genes upon FDI-6 treatment with FOXM1-bound promoters) are most enriched. (**F**) Fraction of genes whose expression levels are changed upon FDI-6 treatment and involved in each cell cycle phase are shown.

## Data Availability

The ChIP-seq and RNA-seq data generated in this study are accessible through GEO accession number GSE201272. Additional data supporting our findings are available from the corresponding author upon reasonable request.

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
