# Peer review of "Characterizing and Targeting Genes Regulated by Transcription Factor MYBL2 in Lung Adenocarcinoma Cells"

_cancers, 2022, doi:10.3390/cancers14204979_

Round 1
Reviewer 1 Report
Review report is attached

Reviewer 2 Report
This manuscript by Lee et al presents data regarding genes regulated by the transcription factor MYBL2 in lung adenocarcinoma cells. The authors use ChIP-Seq to show that MYBL2 mainly binds promotors of genes highly expressed in in lung adenocarcinoma cells and show many of these sites are also bound by FOXM1. They go on to identify FOXM1 and MYBL2 target genes through RNA-Seq and finally present survival data obtained using FDI-6, a FOXM1 inhibitor. This work is interesting and provides new mechanistic insight into the role of MYBL2 in lung adenocarcinoma, however some additional experiments would be required to strengthen some of the claims made in the manuscript. My particular concern is that the experiments with FDI-6 have not been performed rigorously enough to justify the legend for Figure 5 ‘FDI-6 inhibits the activities of FOXM1 and MYBL2.’
Major points
1) In Figure 1 the authors present ChIP-Seq in A549 cells, which they have used because they have high levels of MYBL2 expression (Figure 1A). I would like to see some ChIP-seq data in lung adenocarcinoma cells expressing low or lower levels of MYBL2 (for example AG04450) to determine if all binding sites are still present or if MYBL2 binding is redistributed when it is no longer overexpressed. This would determine the relevance of MYBL2 binding in MYBL2-low vs MYBL2-high lung adenocarcinomas.
2) Overall Figure 5 needs more work. I feel the cell counts (Figure 5A) should be repeated using siRNA for FOXM1 to compare how much of an effect the inhibitor is having vs. a complete knockdown. Also, it would be interesting to see how FDI-6 affects the viability of lung adenocarcinoma cells with lower MYBL2 levels, would you expect them to be less/more sensitive? The colony formation assay (Figure 5B) gives very limited information with only two concentrations of inhibitor. Was this only done once? The legend does not specify. The untreated A549 representative image is not good. It would be better to optimise the cell numbers to allow counting of colony number to quantify the effect of the inhibitor. I would suggest as previous work in MDA-MB-231 cells (Ulhaka et al 2021) has used concentrations ranging from 0.3-20µM and observed a 40% decrease in cell viability with the lowest concentration to try these. Finally, the authors show that FDI-6 reduces protein expression of FOXM1 and MYBL2 (Figure 5C). Similarly to my first point, how does this reduction in protein expression in the presence of FDI-6 affect promotor binding as presented in Figure 1?
3) The paper is focussed on the role of FOXM1 and MYBL2 in activation of cell cycle-related genes. I would like to see a cell cycle profile in these cells with MYBL2 or FOXM1 knock down (siRNA or inhibitor) using BrdU or Ki67 staining for example, to determine exactly how the cell cycle is being affected.
Minor points
1) Some figures are labelled ‘siBoth’ to indicate the use of siMYBL2 and siFOXM1. I think this would be better for the reader if it was specified as siMYBL2/FOXM1 for clarification.
2) There are some instances in the main results text where too much information regarding the specific methodology is given. For example, line 304 ‘FOXM1 antibodies available for purchase…. Cat numbers etc.’ I think this type of detail is better off in the methods section. Also line 307, ‘identifying two biological replicates that passed QC.’ This is important information but again this should either be put in the methods section or included in the figure legend.
3) For Figure 5C, which cell line was this western blot done in – A549 or NCI-H2126? This is missing from the legend.
Reviewer 3 Report
A study by Lee et al investigates the role of transcription factor MYBL2 in lung adenocarcinoma cells. The study involves a complex bioinformatic analysis as well as experimental validation performed in vitro.
Specific comments:
1. Please create a separate paragraph in Materials&Methods for statistical analysis.
2. Fig. 3b and further figures/descriptions - please change "siBoth" to specific names of transcripts targeted by siRNA.
3. Please review English throughout the manuscript especially with regard to unclear statements.
4. Fig. 5 - please correct the units of concentration.
5. Fig. 5A - "hours" below the graphs should be replaced with "time".
6. FDI-6 shows relatively high activity already at 10 uM (Fig. 5). Have you tested lower concentrations of this compound?
7. Fig. 5C - b-actin is not an appropriate marker for nuclear fraction. Please stain another one e.g., lamin B.
8. Please discuss the results in light of more current literature.
Reviewer 4 Report
Yuri Lee et. al tried to characterize the genes that regulated by transcription factor MYBL2 and reveal its molecular mechanisms in lung adenocarcinoma cells by performing CHIP-seq and knockdown experiments. The manuscript was well written, however, there are some major questions to be addressed.
Major:
1. The main findings have already been published in the ref 35 (Ahmed, F., Integrated Network Analysis Reveals FOXM1 and MYBL2 as Key Regulators of Cell Proliferation in Non-small 690 Cell Lung Cancer. Frontiers in oncology 2019, 9, 1011.), which identified a local “driver-network” and its upstream regulators responsible for the cell proliferation in NSCLC and proposed that the FOXM1 and MYBL2 could be promising biomarkers and therapeutic targets for NSCLC treatment. Indeed, besides this, the authors of this manuscript also found some other genes that related to the knockdown of MYBL2, such as CENPA, but they did not further reveal the mechanisms of the changes and how these genes connected. The novelty of this manuscript was not prominent with the current data.
2. The authors mentioned that FDI-6 was a FOXM1-specific inhibitor, however, with the treatment of FDI-6, both the FOXM1 and MYBL2 were significantly downregulated. What’s the mechanism of the reduction of MYBL2? If MYBL2 will be downregulated automatically with the inhibition of FOXM1, why FOXM1 knocking down did not show the same effect (Fig S3)?
Minor:
There are many typos/errors in the figures, for example, no statistical significance in Fig 1A, the y-axis label of Fig 4C, and the concentration unit symbol in Fig 5A, etc. The authors should check the manuscript carefully and correct them.
Reviewer 5 Report
The manuscript by Lee et al-- Characterizing and targeting genes regulated by transcription factor MYBL2 in lung adenocarcinoma cells, describes the oncogenic role of MYBL2 in regulating gene expression in lung adenocarcinomas.
As demonstrated in the study, the manuscript is conceptually very well presented, with adequate experimental evidence and correlative inferences obtained from functional genomics and cell culture approaches.
Although the manuscript appears to have a substantial influence, the authors should provide a few more outcomes to support the significance of MYBL2 in lung adenocarcinomas.
Given below are my concerns:
(1) Authors should mention the source of normal alveolar epithelial cells (AEC) and airway epithelial cells, and their culture conditions. Also, the source of IMR90 is not mentioned in the material-method section.
(2) In figure-1, the authors compare the expression of MYBL2 in normal epithelial and one lung adenocarcinoma cell line, A549. To extend this conclusion in general, the authors should include additional lung adenocarcinoma cell lines for expression analysis.
(3) What conclusions do the authors draw from the ChIP seq genome browser tracks shown in Fig 1B, are not clear, and not discussed in the results? There must be a logical reason to show tracks, and what targets are been referred to here. Fig.C-F well summarizes the overall output of Chipseq data, and I think Fig. 1B can be omitted, as it creates confusion.
(4) Fig. 1G, ChipSeq tracks show MYBL2 peaks, it would be nice to include or merge H3K4me3, H3K27ac, and CTCF tracks here, that makes more sense here than in Fig.1.
(5) Figure-5 A and B, there is a typo in concentration.
(6) Authors claim that FDI-6 alters cell proliferation and inhibits the activities of FOXM1 and MYBL2 in lung adenocarcinoma cells based on in-vitro data. There are two suggestions for this, (i) FDI-6 concentration used for in-vitro experiment is physiologically very high, lower concentration of the drug should be tested, and determine IC50 of FDI-6 in the cell lines tested. Cell killing could be due to the off-target cytotoxic effect of the drug also, (ii) It would be great if authors demonstrate the effect of FDI-6 on in-vivo tumor growth in mice using A549 xenografts. This will help the authors to establish the claim that FOXOM1 inhibition with FDI-16 has anticancer effects in lung adenocarcinoma
(7) The authors make a big claim about MYBL2 oncogenic role, but it is not directly investigated in the current manuscript, and the authors largely focus on the role of MYBL2 target genes. It would be worth doing an in-vivo mouse tumor xenograft experiment in wild-type and MYBL2 knockdown A549 cells, which will help establish the oncogenic role of MYBL2 in lung adenocarcinoma.
Round 2
Reviewer 1 Report
Authors have addressed all the concerns. The study is novel with interesting findings. Congratulations on your work.
Reviewer 2 Report
I had three major points regarding this manuscript and only one has been partially addressed. As far as I can tell the only new data which has been provided is some FOXM1 ChIP-qPCR in the presence of FDI-6. This is interesting and indicates that the inhibitor affects FOXM1 binding to important genes such as CENPA. However, I feel the minimum that should be done to complete this paper should be:
1) Figure 5A - the authors indicate they have performed the experiment with FOXM1 siRNA and it is difficult to compare the data to the inhibitor studies due to variations in cell numbers. Could this data be presented in % form instead of actual cell numbers to get around this?
2) I would still like a see a cell cycle profile in these cells when siRNA for MYBL2 and/or FOXM1 is used. The paper is very cell cycle focused and determining the functional significance of changes in cell cycle-related genes is critical to the message.
Reviewer 3 Report
The Authors have sufficiently addressed my previous comments. Please correct Fig. 4C that still contains 'siBoth'.
Reviewer 4 Report
The authors have addressed most of my questions/concerns. However, the FOXM1 and MYBL2 as potential biomarkers and targets were studied and published by others. The authors should include a paragraph in discussion section to highlight the novelty of their manuscript and discuss the caveat of their study.
Reviewer 5 Report
The authors have addressed the concerns.
Round 3
Reviewer 2 Report
The authors have addressed my comments and I feel the manuscript has been sufficiently improved for publication.